# First-Hand Experience of Severe Dysphagia Following Brainstem Stroke: Two Qualitative Cases

**DOI:** 10.3390/geriatrics5010015

**Published:** 2020-03-04

**Authors:** Annette Kjaersgaard, Hanne Pallesen

**Affiliations:** Hammel Neurorehabilitation Centre and University Research Clinic, Aarhus University, 8450 Hammel, Denmark; hannpall@rm.dk

**Keywords:** dysphagia, rehabilitation, neurology, qualitative interview, phenomenological design

## Abstract

Background: Dysphagia has profound effects on individuals, and living with dysphagia is a complex phenomenon that touches essential areas of life. Dysphagia following a brainstem stroke is often more severe and the chances of spontaneous recovery are less likely as compared with dysphagia following a hemispheric stroke. Objective: To explore how two individuals with brainstem stroke experienced severe dysphagia during their inpatient neurorehabilitation and how they experienced their recovery approximately one month following discharge. Methods: An explorative study was conducted to evaluate the first-hand perspective on severe eating difficulties. A qualitative case study was chosen to collect data during two face-to-face semi-structured interviews. Phenomenological perspectives shaped the interview-process and the processing of data. Results: Analysis of the empirical data generated the following main themes regarding experiences of: (i) the mouth and throat; (ii) shared dining; and (iii) recovery and regression related to swallowing-eating-drinking. Conclusion: Participants expressed altered sensations of the mouth and throat, which affected their oral intake and social participation in meals. Good support for managing and adapting their problems of swallowing, eating, and drinking in daily activities is essential. Knowledge and skills of professionals in relation to dysphagia is a significant requirement for recovery progress in settings within the municipality.

## 1. Introduction

About 15% of patients in neurorehabilitation are diagnosed with brainstem stroke (BSS) [1]. The swallowing centre or the “central pattern generator for swallowing” (CPG) is located in the brainstem [2]. Studies report an incidence of dysphagia in 70–81% of patients with BSS at admission to acute stroke unit [3,4]. Teasell et al. reported an incidence of dysphagia in 47% of the patients with BSS admitted to neurorehabilitation, and found that patients with BSS experienced a range of reduced functional abilities e.g., hemiparesis, dysarthria and dysphagia, diplopia, and ataxia as a consequence of stroke [1]. Although the occurrence of dysphagia appears to be fairly high among patients with BSS and they often require enteral tube feeding in the acute phase of rehabilitation, they are found to have a better long-term prognosis than patients with hemispheric stroke [5]. 

Dysphagia has profound effects on individuals [6], and living with dysphagia is a complex phenomenon that touches essential areas of life [7] and is common during recovery from stroke in the community [8]. Individuals who experience dysphagia as a result of stroke have a complex rehabilitation journey involving the interaction of many physical, emotional, and social considerations. Multiple affecting factors have been described, including meaning-related, functional, and contextual factors, and the consequences of non-intervention, feelings of loss, social isolation, negative sensations, and depression—all of which can potentially increase the impact of disease severity, functional outcome, and quality of life in patients with stroke [6]. Healthcare professionals should not only be aware of the physical but also the significant psychosocial consequences of living with dysphagia following a stroke [9]. Contrasting opinions on the psychological issues of dysphagia have been identified between acute and chronic stroke patients, which differ from the perspectives of clinicians and caregivers [10]. It has been highlighted that it is important for treating clinicians to be aware of psychological issues, to address them according to the patient’s clinical recovery, and to consider the interplay between psychological and biomedical consequences. Findings of another study indicate that individuals with post-stroke dysphagia experience a lack of support from healthcare professionals, and concluded that better healthcare support following discharge from hospital is required to ensure optimal quality of life [11].

Dysphagia can impact social opportunities and the pleasure derived from meals and the activity of eating and feeding, as well as the quality of social relationships for the individual with BSS, undermining their health and confidence. Individuals with dysphagia may become isolated, feel excluded by others, and be anxious and distressed at mealtimes [12], often experiencing considerable limitations in their everyday life. The complexity of experiencing eating difficulties following a stroke and issues related to feeling in control are based on a number of strategies: being careful when eating, avoiding social activities, needing help from others, analysing the consequences of eating different foods, and eating safely and properly [13]. A central theory for understanding the dining experience of individuals with dysphagia living in care facilities that has emerged across all participant accounts is focusing on the individual: individualisation and socialisation are at the heart of mealtime care [14]. 

In Denmark, it is standard practice to screen the swallowing function within 24 h of admission to an acute stroke unit. Over the last decade, assessment and treatment of patients with dysphagia and related activity problems has become an occupational therapy practice and specialist field in Denmark. The occupational therapist is central to the interdisciplinary team for the treatment of patients with dysphagia and focuses on the holistic perspective of oral intake. Dysphagia and related activity problems are seen not only as medical issues but also cognitive and social components that can affect patient prerequisites for successful swallowing-eating-drinking. When activity problems related to swallowing-eating-drinking occur, it is an important part of the essential occupational therapy focus area [15].

Despite the fact that there are a significant number of patients with BSS admitted to inpatient neurorehabilitation, there is little research published on this population, unlike rehabilitation for patients with hemispheric stroke, for which there is a relatively comprehensive research literature base [1]. It has been recommended that on-going research in the area of stroke rehabilitation needs to include an emphasis on exploring the experiences of stroke survivors [16]. Little is known regarding the lived experience of dysphagia following BSS from a patient perspective [9], which is significant knowledge for daily practice in occupational therapy in order to provide the best assessment and treatment of individuals with dysphagia. 

The aim of the present case study was to explore how two individuals with BSS experienced severe dysphagia during their inpatient neurorehabilitation from a patient perspective and how they expressed their recovery approximately one month following discharge.

## 2. Materials and Methods 

A qualitative case study was conducted according to the methodology described by Yin, 2009 [17]. According to Yin [17]

“Case studies are the preferred method when (a) “how” or “why” questions are being posed, (b) the investigator has little control over events, and (c) the focus is on a contemporary phenomenon within a real-life context”.

Phenomenological perspectives shaped the interview process and the processing of data. Inspired by phenomenology [18,19], the body is situated in a dynamic living world, understood as the participant’s past, future, human attitude, physical, ideological, and moral situation, embodied as well as connected to the world [18,19].

The CARE Case Report Guideline [20] was used to increase the accuracy and transparency of the present case report study, which forms one of four sub-studies of a mixed-methods investigation regarding difficulties with swallowing and eating following acquired brain injury (ABI). The first of these studies was a prospective randomised controlled trial (RCT) of the assessment of facial-oral tract therapy versus fibreoptic endoscopic evaluation of swallowing during inpatient neurorehabilitation; we compared the risk of aspiration pneumonia in patients with ABI [21] and the time to initiation of oral intake and recovery of total oral intake prior to discharge [22]. The second study, which has been previously published, was a preliminary, explorative, qualitative interview conducted with a view to gathering patient perspectives on eating to refine the methodology for a later longitudinal study [23]. 

### 2.1. Sampling and Participants

The two participants were recruited through a criterion sampling strategy [24]. The inclusion criteria for the present pilot study were: (1) diagnosed with an BSS and enrolled in the above-mentioned study; (2) severe dysphagia at the time of admission to inpatient neurorehabilitation (Functional Independence Measure (FIM) score of 1 for the item “Eating”); (3) have or had a feeding tube; and (4) able to understand the interview questions and express/describe their experience in Danish (FIM score 5–7 for items “Expression” and “Memory” at the time of discharge from neurorehabilitation). The participants in this study were retrospectively selected, with help from local clinical dysphagia experts. Three patients of a total 119 patients in the RCT [21] were diagnosed with BSS, however only two of those fulfilled the inclusion criteria. Only patients with BSS were chosen because of the severity of their dysphagia. Their need of feeding tube and their appropriate cognition and memory made them able to express their first-hand experiences. 

### 2.2. Ethics

The present case study was performed according to the Declaration of Helsinki [25]. The Danish Data Protection Agency was notified, and the collection of data was handled according to their guidelines (Journal no. 2007-58-0010). The participants gave their verbal and written informed consent for participation and were guaranteed confidentiality. Participation was voluntary, and the participants could withdraw from the study at any time. The data were de-identified and pseudonym names were created. 

### 2.3. Data Collection

The quantitative data are from the RCT [21], which were documented from the medical records by the treating occupational therapist. These data were used to describe patient characteristics and contextualise the participants in the present case study. The Functional Independence Measure (FIM) [26] was rated by the multidisciplinary team as standard practice at the clinic. 

The empirical data were collected using semi-structured interviews. An interview guide was used during the interviews, which (Table 1) consisted of topics derived from the literature [27]. The questions were open-ended [28] and developed following a pilot interview with an individual with ABI, during which the need to simplify the terminology and prompts was identified. 

The participants were interviewed once during inpatient neurorehabilitation and once in their own homes one month following injury.

During the interviews, the participants were encouraged to give examples and describe actual situations that they had experienced [29], and they were asked to describe their present and previous experiences and management of eating difficulties. 

The first author performed all the interviews, which lasted 30–60 min and were tape-recorded and fully transcribed verbatim by an individual not involved in the present study. 

### 2.4. Data Analysis

The data were analysed using a five-step phenomenological analysis, which enables the researchers to enter the world of the participants and gain insight into their thoughts, feelings, and experiences [29,30]. 

Validation in qualitative research is a process of continually checking, questioning, and theoretically interpreting the findings [31] to secure trustworthiness [32]. The step for improving credibility included careful efforts to ensure that the speech of the participants was properly understood. Following transcription of the interview, the first author returned the transcripts to the participants to verify that the researcher had correctly understood their statements. The researcher triangulation was undertaken by two skilled researchers and involved the extraction of meaning units, formulation of themes, interpretation of these data, and use of adaptation theories to discuss the findings. Triangulation was performed to increase the credibility of the present study (Table 2).

## 3. Results

The analysis of the quantitative data is shown in Figure 1 and Figure 2. The timelines illustrate the participants’ relevant past medical history, current illness, functional and social abilities, physical examination, diagnostic evaluation, interventions, and rehabilitation outcome. The participants’ difficulties related to swallowing and eating from the time of BSS to one month following discharge from inpatient neurorehabilitation are described.

The analysis of the empirical data generated the following main themes regarding experiences of: (i) the mouth and throat; (ii) shared dining; and (iii) recovery and regression related to swallowing-eating-drinking. 

This section describes the central phenomena that the patients experienced during the recovery process in relation to their mouth and throat and difficulties in swallowing and eating following BSS. Furthermore, the section describes the significance of reduced sensation in the mouth and being forced to obtain food through a feeding tube. A discussion of how these experiences affected the patients’ social life later in the recovery process is included.

### 3.1. Mouth and Throat

#### 3.1.1. Unusual Sensations and Feelings

The two patients described how they constantly had a foreign experience with respect to their mouth and throat during the early rehabilitation phase in the hospital. They had too much saliva, which they could not swallow, and at the same time, had a need to clear their throat. Prior to BSS, swallowing and cleaning the mouth and throat had been a normal and effortless task without conscious consideration. Following BSS and during the early stage of rehabilitation, their energy and attention were often focused on difficulties in swallowing, coughing, and voice clearing.

“Clearing my voice ... I often did this ... and saliva was quite crazy in the beginning ... And then I couldn’t swallow either the mucus or the saliva. It all had to come out. It was damn annoying ... When it (the saliva) came, I spit it out. It’s rare that it all comes up”.(Ole)

Several times during the interview, talking triggered a coughing attack. 

“I think it’s disgusting when I start coughing in the middle of everything. I mostly eat by myself”.(Bo)

The risk of a coughing attack was one of the main reasons why Bo chose to eat alone. He also had problems with clearing his voice, and the volume of his voice had become weaker, which Bo also found monotonous.

“Those who know me do understand what I am saying. But my voice, it has become more monotonous than it was before”.(Bo)

#### 3.1.2. The Pleasure of Having Something Familiar in the Mouth

Both patients explained that chewing, swallowing, and eating food were not things that they had thought about prior to BSS. During the early stage of the rehabilitation process, the matter of not being able to chew, swallow, or eat became apparent, but the absence of food in the mouth and instead obtaining food through a feeding tube was difficult to acknowledge and accept. However, when they were allowed to regain different tastes and sensations of food, they felt great pleasure in having something familiar in the mouth and recognised the positive experience of taste.

“The first portion of yogurt was like getting a better Christmas dinner”.(Bo)

Ole expressed, while still having the feeding tube and not being allowed to eat or drink, that he has enjoyed the taste of a familiar soft drink. 

“I haven’t tasted any proper food since I had the stroke … I rinse my mouth with soda and such like. It seems to help”.(Ole)

Bo expressed, after leaving the hospital and being at home, that he had experienced a change in taste, since prior to the BSS he had enjoyed drinking beer. But following BSS:
“It (the beer) no longer tastes good because it tastes like a “lazy beer” (without gas) when it is turning inside the mouth .... that is the same with red wine. The quality of the red wine has to be much better today than before, before I liked the taste of it”.(Bo)

Bo acknowledged that he had to keep the food or drink longer in his mouth before he could swallow it; therefore, he believed that he tasted more of the beer than he did prior to BSS, which may be the reason why the quality had to be higher.

#### 3.1.3. Unfamiliar Objects “Invade” the Mouth

As part of the rehabilitation, both patients received oral stimulation as a treatment intervention, during which the occupational therapist stimulated the gums, cheeks, and tongue using her gloved finger to stimulate sensation, movement and facilitate the swallowing of saliva [33]. 

Ole received chewing training, during which he had to chew on a piece of apple wrapped in gauze, because he was not able to eat food of harder consistencies [33]. He made the following comment:
“I’ve tried that, but I don’t like it. They (therapists) put something in the gauze that I have to chew on and that I do not like. That’s the gauze I can’t have in my mouth, it’s something strange....”.(Ole)

Subsequently, Ole expressed that he has chewed chicken, which after a while, he had to spit out.

“It was fried well, and it was just before it melted on the tongue. But chewing in gauze gives me a feeling of nausea”.(Ole)

Ole had ambivalent feelings towards oral stimulation. He acknowledged that there was a reasonable explanation why it was a good treatment for him; however, the negative part was:
“As a therapy, it is not horrible. But I can just taste their (therapists) rubber gloves and those tasted nasty”.(Ole)

Ole believed that he did not produce more saliva during oral stimulation and felt that this treatment made no positive changes to the way he swallowed saliva.

### 3.2. Shared Dining

Both patients recognised that a shared meal was important to them for maintaining, establishing, and developing social contacts. However, during their hospital stay, while they had a feeding tube and were not allowed to eat or drink regular food, they both chose not to participate in the rehabilitation clinic’s shared meals in the dining room. Even though they themselves decided not to join the shared meals, they also expressed that it was annoying and frustrating to not be able to sit down and have a good time with peers while eating. “It’s half our life”. Bo did not want his peers to see him during a coughing attack. Furthermore, he did not think he belonged there while he was not able to eat with them … “it seems odd just looking at them and how they enjoy their meal without having anything myself”

#### 3.2.1. Eating Together

Enjoying a meal with others and communicating with each other is a well-known Scandinavian tradition, which was also appreciated by the two participants. However, this particular practice had changed since BSS and remained a problem following discharge from the hospital. Bo explained: 

“I need to take my time and I have to concentrate on eating …… my wife quickly got used to it, so she knows very well that she should not talk to me when I have food in my mouth ... but it’s so annoying when you’re seeing other people. You may well be perceived as a little ignorant”.(Bo)

Bo described that he had explained to his friends that he has difficulties in eating, chewing, and swallowing, and how these difficulties affected him. He also recognised that he avoided shared dining with people he did not know very well, which had reduced his social life.

#### 3.2.2. Social Interaction

Bo’s eating difficulties had minimised his social activities, and he highlighted that he carefully chooses which social events to participate in. 

“I have to concentrate when eating so it’s difficult to have a conversation at the same time… it’s very annoying when I eat with other people…. And others perceive me as snobbish”(Bo)

Ole, on the other hand, after short stays at a regional neurorehabilitation unit and an acute infection unit, was admitted to a nursing home. His functional level had dropped considerably, and he spent all of his time alone in his room, mostly in bed.

### 3.3. Recovery and Regression

Despite Bo’s difficulties related to eating having reduced his social life, he also described how he had adapted to these difficulties and that he was still able to live a meaningful and valued life with his wife, family, and friends. Bo often used the term “I have learned to live with…”. He had discovered and invented new strategies to overcome his difficulties: 

“I do not talk with food in my mouth, and the meals at home have got another rhythm - eating and talking are separated”.(Bo)

But for Ole, the situation following discharge from the hospital was completely different. He had experienced no progress at any functional levels. His difficulties in eating had increased significantly and he had received a feeding tube again after a pneumonia. He explained:
“I’m not getting well by just lying down”.(Ole)

Ole seemed fully aware that his situation is serious and that he is about to give up. 

“My health is miserable… the damned pneumonia… it fills my head that I can’t get anything to eat and drink… it’s damn hot outside right now”(Ole)

Ole also described that he did not get the support that he needed, and he seemed strongly pessimistic and discouraged during the second interview, during which he was living in a nursing home. His voice was almost absent, and during the interview he was whispering and needed many breaks. As illustrated in Ole’s timeline (Figure 2), the functional oral intake scale (FOIS) was 1, indicating that he had no oral intake and received all his nutrition via a feeding tube. He was also receiving treatment for pneumonia. His condition had declined and was alarming. Unfortunately, the staff’s knowledge of dysphagia in the nursing home seemed very limited, and it appeared that staff had accepted the regression and not asked for professional expertise and support with respect to the worsening in his functional level and reduced quality of life.

## 4. Discussion

The aim of this study was to explore how two individuals with brainstem stroke experienced severe dysphagia during their inpatient neurorehabilitation and how they experienced their recovery approximately one month following discharge. The two cases described in the present study illustrate that the mouth and throat are essential areas related to ingestion, social dining, and communication. Having dysphagia, leading to difficulties in ingestion, swallowing, eating, and drinking, had affected their awareness of the mouth and throat, sensation, taste, and feelings, and caused different tastes and consistencies of food, drink, and objects in the mouth. Furthermore, dysphagia had reduced trust in their own ability to control eating and drinking, leading to mealtimes in solitude and reduced social engagements.

Dysphagia is not a medical diagnosis but rather a dysfunction, and is therefore described by its symptoms, clinical signs or physiologic impairments [34]. The two participants in the present study described strangeness and an odd sensation and feeling in the mouth and throat, which were present with or without having any objects or food in the mouth. Particularly, during the early stage of rehabilitation, they felt that they constantly had too much saliva that they could not swallow or remove from the throat. The participants could not control this particular area of their body in the same way as they could prior to BSS; they strove for control and had to be careful while eating and swallowing. These basic bodily functions were no longer automatic, and they struggled to comprehend the enormity of what had happened. Furthermore, during the early rehabilitation stage, they had a feeding tube due to the risk of aspiration, and thus were unable to swallow harder types of food. The joy of tasting food and the feeling of satiation is one of the most important aspects of well-being [35]. However, while striving for regaining control of the throat, both participants avoided settings that involved shared meals, and thereby excluded themselves from an important part of their social life. The fear of a coughing attack was an explanation given by the participants. Feelings of shame, worry, and panic have been described in other studies [9,23].

Dysphagia appears to lead to changes related to their body and environment. The patients can no longer ingest food as they once did nor participate in meals the way they did prior to BSS. As described by Merleau-Ponty [18], the patients do not participate in the world in the same unpredictable and self-evident way as they did before. 

Another phenomenologist refers to bodily capacity and the sub-terms, body scheme and body image, which not only deal with motor skills but also include an understanding of one’s own body, conceptual understanding of the body generally, and attitudes and settings of one’s own body, including belief in one’s own ability [36]. These aspects appear to be important during the process of developing a realistic belief in one’s own skills and confidence in overcoming various barriers, and thus mastering the new life situation [37,38]. Using these theoretical concepts to frame the empirical material of the present study, it can be assumed that patients’ changed sensory motor function related to the mouth and throat not only affects the patients’ perception of body scheme but also body image. Usually, we do not think about how we chew food and we are not aware of swallowing food before we talk. The body, and in this case the mouth and throat, are almost present as an invisible curtain for our actions. Many researchers have described how the body, due to illness or disability, becomes an obstacle to living as desired. Our “I can” disappears and we are disturbed in the projects that define “who we are” [36,39,40].

Although it makes sense to separate the body scheme and body image conceptually, the systems are closely related to targeted behaviour and intentional action. They encompass a dynamic system that changes over time and in different practices that require different attention; however, the body scheme works at a higher level (most optimal) when attention is directed away from the body [41]. Health professionals who support an individual during the rehabilitation process can choose an approach with more focus on one of the systems by creating different frames; either a more introverted focus on the body or participation in interaction with others, where one has to control one’s own body but also direct attention towards the surroundings. Data generated by the present case study does not tell us which approach the health professional used or whether they focused on the concepts of body scheme or body image. We can, however, postulate that there was more focus on the mouth, throat, and dysphagia during hospital-based inpatient rehabilitation than in the municipal nursing home.

Many conditions are important for recovery from an illness, and even if individuals have experienced the same type of stroke, it does not mean that they will recover to the same extent, despite receiving the same support and interventions. The severity of BSS has a major impact on recovery; Ole was more severely physically impaired than Bo. Another important aspect concerning recovery is the health professionals’ knowledge, capabilities, and conceptual understanding of the practical execution to perform intervention that leads to recovery, looking for even discrete turning points [42]. Several studies have found that the trajectory of stroke recovery may necessitate a re-evaluation of life plans, a rethinking of priorities, and an integration of resulting disabilities into current and emerging life situations for both stroke survivors and their caregivers [43,44,45]. Bo’s recovery appears to follow these examples and intentions. However, for Ole, there was clear progress while he was in the hospital and had access to experts’ knowledge related to dysphagia; unfortunately, this situation was different in the nursing home, since health professionals’ support and interventions seem to be absent.

It seems obvious that stroke rehabilitation, its approach, and interaction with the patient and relatives should be based on recovery. Every individual with stroke should be involved in a process of re-defining themselves regarding re-integration into the community and have the possibility of being supported by professionals during this transition, even when living in a nursing home. 

### 4.1. Limitations

Due to the small sample size of only two participants, the conclusions and generalisation with respect to the BSS population are limited. It is also important to consider that the inclusion of participants until saturation may have added additional experiences, findings, and conclusions. 

### 4.2. Future Research

Future research is required, and it appears that important knowledge of clinical relevance on this topic is lacking, particularly concerning long-term follow-up from a patient perspective of the experience of severe dysphagia several years following BSS. The recommendation for future research is to explore and interpret how individuals with BSS and their caregivers experience and adapt to problems and recovery over time. Furthermore, it is important to verify whether direct interventions of information, education, and psychological support on this topic can improve patient well-being and make the overall rehabilitation program more effective in both a hospital setting and nursing homes or specialised institutions for individuals with ABI.

## 5. Conclusions

The two participants with brainstem infarction and severe dysphagia expressed altered movement and sensation of the mouth and throat, affecting both their oral intake of food and drink and their social participation in meals. During their inpatient hospital stay, they both received good support for the management and adaptation of their problems related to swallowing, eating, and drinking in daily activities. Following discharge, their recovery processes were described quite differently. The experiences expressed in the two present case reports could indicate a lack of knowledge and skills of professionals in relation to dysphagia, which might be significant requirements for recovery progress in some settings within the municipality. Further clarification of a lack of knowledge and skills is needed. The findings from this study provide a preliminary step in understanding the BSS patient’s perspective of severe dysphagia during the first four to six months post stroke.

## Figures and Tables

**Figure 1 geriatrics-05-00015-f001:**
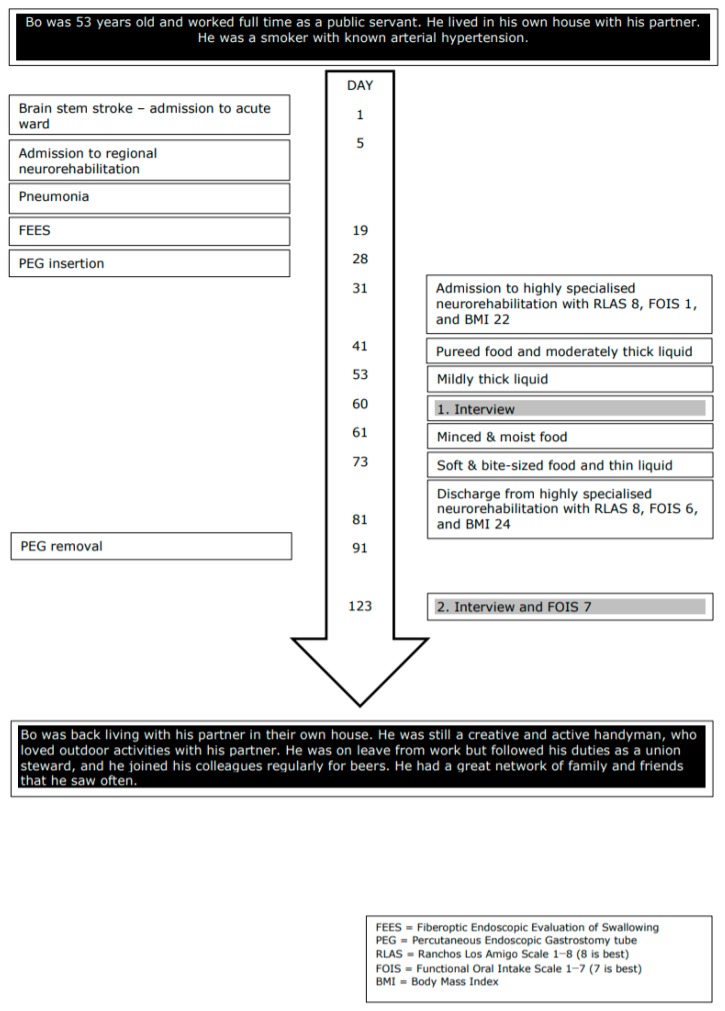
Timeline Bo.

**Figure 2 geriatrics-05-00015-f002:**
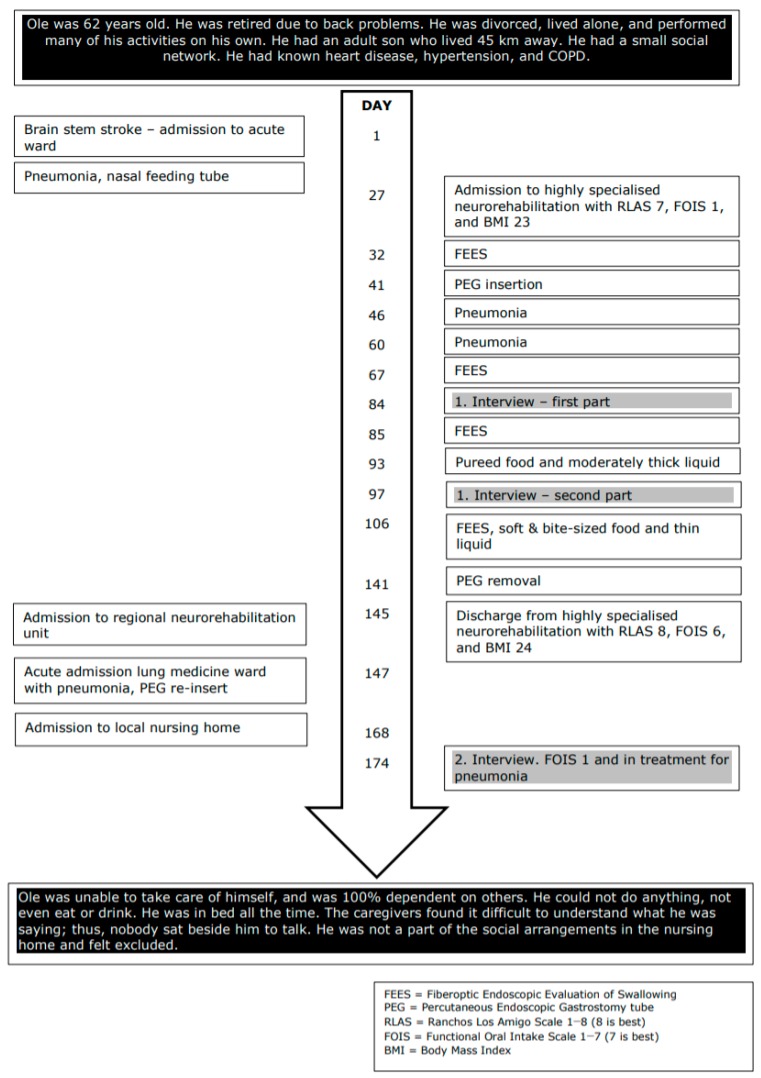
Timeline Ole.

**Table 1 geriatrics-05-00015-t001:** Domains in the semi-structured interview guide.

Introduction
1. General questions related to eating and drinking 2. The meaning of food and liquid prior to the injury 3. The meaning of food and liquid at the time of the interview and immediately after the injury 4. Are you currently experiencing any physical difficulties that may influence eating and drinking? How was it immediately after the injury? 5. Are you currently experiencing worries and if so, do they influence your mood in relation to eating and drinking? How was it immediately after the injury? 6. Your social life (meals with the family, work, leisure activities, parties, vacations, etc.)—how is it currently? How was it immediately after the injury? 7. Your experiences of obtaining food and drink via a feeding tube Closing interview (debriefing)

**Table 2 geriatrics-05-00015-t002:** Five steps in the analytical process inspired by Giorgi [29].

	Step	Analytical Process
1	The entire interview was read and reread to gain an overall picture	The first author transcribed each interview. Each interview transcript was then read by the other author, after which the interview was played again to ensure that the transcription was accurate
2	Natural “meaning units,” as they were expressed by the interviewees were identified by the researcher	The data were analysed in depth using a phenomenological method to trace thematic patterns of how the two informants experienced severe dysphagia during their inpatient neurorehabilitation and how they had recovered during the one month since discharge. This part of the analysis was first performed by both authors individually and subsequently by consensus of the two authors
3	The dominating themes in the meaning units were identified. The researcher attempted to form themes from the interviewed person’s point of view, as the researcher understood it	Meaning units were organized and gradually transformed into categories. Firstly, the data were separated for each informant, and secondly, similarities and differences were noted in an iterative process
4	The meaning units were questioned based on the research questions from the semi-structured interview guide	The data were described in a final set of themes and sub-themes that answered the research questions regarding how the two people with BSS experienced severe dysphagia during their inpatient neurorehabilitation and how they expressed their recovery approximately one month following discharge
5	The non-redundant themes were condensed into descriptive statements	The first draft of the final results was co-generated and discussed by both authors, after which it was considered to constitute essential knowledge

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
