# Peer review of "First-Hand Experience of Severe Dysphagia Following Brainstem Stroke: Two Qualitative Cases"

_geriatrics, 2020, doi:10.3390/geriatrics5010015_

Round 1
Reviewer 1 Report
This is a well-written manuscript that aimed to evaluate the personal experiences of dysphagia following brainstem stroke in two patients. The authors explored themes relating to the oral and pharyngeal experiences, the social impacts of the dysphagia, and how their dysphagia has changed over time. Comments are below:
Line 106: BSS is written as SSB Were there only 2 patients who fit the criteria? If not, what was the rationale for only selecting 2? In Table 2, there is reference to “research questions,” but it is unclear what these are. Figures 1 and 2 are very helpful. Is it not standard practice to evaluate swallowing function following brainstem stroke in Denmark? It appears that both patients suffered from pneumonia prior to receiving swallowing evaluation. With qualitative research, one of the goals can be hypothesis-generating, resulting in development of new ideas that can be tested in the future. Was that one of the goals of the present study? If so, the discussion would benefit of a further exploration.Author Response
Please see the attachment

Reviewer 2 Report
Introduction:
In general, there are numerous instances where the statement provided in the text is not supported by an appropriate reference/citation. For example, in the first sentence (Line 29) the authors state that “up to 25% of all strokes are lesions affecting the brainstem region”, however, the citation provided is from the article “Beneficial effects of postural intervention on prehensile action 390 for an individual with ataxia resulting from brainstem stroke”. When citing, please provide the original article that supports the statement, merely citing an article that makes that statement but references another article is not sufficient, or appropriate. In this instance, citing the epidemiological study that shows 25% of stroke lesions involve the brainstem is needed.
Line 30: “An estimated 15% of all 30 patients admitted to neurorehabilitation have experienced brainstem stroke (BSS)”, this sentence is directly copy/pasted from the abstract of an article from Topics in Stroke Rehabilitation: The rehabilitation of patients recovering from brainstem strokes: case studies and clinical considerations. This article is not referenced with this statement, please be wary of plagiarism and using appropriate citations.
Line 32: Authors note that “dysphagia following BSS is often more severe and the chances of spontaneous recovery are less likely as compared with dysphagia following a hemispheric stroke [4].” However, the study cited did not compare treatment to hemispheric stroke and was a single study of 10 participants who underwent therapy. This study did not aim to test spontaneous recovery. Again, please use appropriate references throughout the introduction to support statements, perhaps several references together are needed to support statements.
Line 36: Please describe what the authors specifically refer to when stating “long term results are favorable”. What outcomes are they referring to?
Of note, many sentences appear to be rephrased or copied from the article “Kruger et al, 2014”. Please be wary of rephrasing from another article, recommend checking references and statements.
Line 37: Authors make the statement that “It has been shown that 88% of patients resume complete oral intake four months 37 following stroke onset”, this sentence appears to make a general statement about patients with brainstem stroke, however, the study referenced was based on verbal reports from phone calls of 27 patients who noted they were fully eating orally. This single study should not be generalized to the population of all patients with brainstem stroke based on the limitations of the study referenced.
Overall, please thoroughly check all articles referenced in the introduction to ensure original article is referenced and is an appropriate citation to back up each statement, may need to include additional references.
Methods:
Line 106: Please define SSB, or did the authors intend BSS?
Line 106: It is unclear why, specifically, brainstem strokes were an inclusion criteria for the aims of the study? Why not any patient with severe dysphagia regardless of stroke location? Further, please provide the rationale for patients needing to have had a feeding tube.
Table 1: The authors selected questions from an already established and validated swallowing quality of life questionnaire (SWAL-QOL), can authors describe why this validated tool was not used and provide rationale for why selected questions were chosen from this tool?
Throughout the case descriptions, the participants are noted to have “severe dysphagia”, however there is no mention of swallowing physiology to support this claim. Diet status (as determined by the FIOS) would not necessarily be indicative of swallowing severity. It would be helpful for the authors to describe the physiologic impairments from the FEES assessment to provide objective evidence of swallowing severity and function. It would be beneficial to relate the swallowing function of each individual to their experiences. As it stands, we really don’t know what swallowing function actually was for either patient aside from the diet recommendation that is subjective. As each participant is describing the sensation in their throat/mouth, comparing their experiences to their unique physiologic deficits as noted in the FEES would be helpful.
Line 298: Authors note “Dysphagia is not a medical diagnosis but rather a dysfunction, and is therefore described by its symptoms or clinical signs [35]” Dysphagia is in fact a medical diagnosis, although it is not a disease within itself, it is the result of a symptom of another disease. The claim that dysphagia is described by its symptoms or clinical signs is inaccurate, as dysphagia is described by its physiologic impairments (i.e. reduced duration of laryngeal vestibule closure, reduced UES opening, prolonged swallowing reaction time, etc). Clinical signs are noted at the bedside and may be used to screen for dysphagia, however, dysphagia diagnosis is made through instrumental evaluations (i.e. FEES, MBS) where physiology can be objective viewed and measured and compared to established normative data to determine impairment.
Results and Conclusions: In general, the authors provide a thorough description of the experiences of two individual patients. However, the authors conclude that “the knowledge and capabilities of professionals in relation to dysphagia are significant requirements for recovery progress”. It is unclear how knowledge of professionals was measured and there doesn’t appear to be any evidence to support that professionals knowledge of patient experience is a “significant requirement”. The relationship between a professional’s knowledge and recovery progress was not measured or reported. The purpose and rationale for reporting this case study remains unclear. While I agree that taking into account the individual needs and wishes of each patient is important, there is no evidence from this paper that correlates the professional’s understanding of the recovery process. Could the authors please clarify in the conclusions how this article contributes to the literature in a novel way?
Reviewer 3 Report
Overall a well written and interesting paper. However, I do have some questions:
Two participants were recruited for this study. The authors state that these participants were retrospectively selected with the help of dysphagia experts from a total pool of 119 candidates. However, I am unsure precisely why these two were chosen. Were all 119 eligible? Were the two chosen because they represented some particular characteristic? Were they chosen because of convenience? Were they the first to consent? The authors mention that the questions were developed following a pilot interview with a participant with ABI. Can this data be incorporated into the study? In the discussion the authors state that the participants struggled to comprehend the enormity of their condition. Are there any quotes that back this point up? (perhaps they can be added to the results section) Some mention is made about perception of body image. Were there any statements made by the patients that suggest that this was the case (even if not directly asked by the authors)? If so it would be useful to include themAuthor Response
Please see the attachment
